# Ibuprofen-Loaded Heparin Modified Thermosensitive Hydrogel for Inhibiting Excessive Inflammation and Promoting Wound Healing

**DOI:** 10.3390/polym12112619

**Published:** 2020-11-06

**Authors:** Abegaz Tizazu Andrgie, Haile Fentahun Darge, Tefera Worku Mekonnen, Yihenew Simegniew Birhan, Endiries Yibru Hanurry, Hsiao-Ying Chou, Chih-Feng Wang, Hsieh-Chih Tsai, Jen Ming Yang, Yen-Hsiang Chang

**Affiliations:** 1Graduate Institute of Applied Science and Technology, National Taiwan University of Science and Technology, Taipei 106, Taiwan; habegaz21@gmail.com (A.T.A.); fentahunhailebdu@gmail.com (H.F.D.); tefe16@gmail.com (T.W.M.); yihenews@gmail.com (Y.S.B.); Endris_Yibru@dmu.edu.et (E.Y.H.); wherelove8@gmail.com (H.-Y.C.); cfwang@mail.ntust.edu.tw (C.-F.W.); 2Advanced Membrane Materials Center, National Taiwan University of Science and Technology, Taipei 106, Taiwan; 3Department of Chemical and Materials Engineering, Chang Gung University, Taoyuan 320-338, Taiwan; cyh4d25@adm.cgmh.org.tw; 4Department of General Dentistry, Chang Gung Memorial Hospital, Taoyuan 320-338, Taiwan

**Keywords:** anti-inflammation, heparin, ibuprofen, thermoresponsive hydrogel, wound healing

## Abstract

Hydrogels have been investigated as ideal biomaterials for wound treatment owing to their ability to form a highly moist environment which accelerates cell migration and tissue regeneration for prompt wound healing. They can also be used as a drug carrier for local delivery, and are able to activate immune cells to enhance wound healing. Here, we developed heparin-conjugated poly(*N*-isopropylacrylamide), an injectable, in situ gel-forming polymer, and evaluated its use in wound healing. Ibuprofen was encapsulated into the hydrogel to help reduce pain and excessive inflammation during healing. In addition to in vitro studies, a BALB/c mice model was used to evaluate its effect on would healing and the secretion of inflammatory mediators. The in vitro assay confirmed that the ibuprofen released from the hydrogel dramatically reduced lipopolysaccharide-induced inflammation by suppressing the production of NO, PGE2 and TNF-α in RAW264.7 macrophages. Moreover, an in vivo wound healing assay was conducted by applying hydrogels to wounds on the backs of mice. The results showed that the ibuprofen-loaded hydrogel improved healing relative to the phosphate buffered saline group. This study indicates that ibuprofen loaded in an injectable hydrogel is a promising candidate for wound healing therapy.

## 1. Introduction

Wound care therapy is a major clinical burden, as it is a complex multi-step process comprising hemostasis, inflammation, angiogenesis, proliferation, connective tissue remodeling, and wound strength recovery [1,2,3]. Wound healing restores the structural integrity of the damaged tissue through spontaneous growth and regeneration [4]. During this process, communication between keratinocytes, fibroblasts, endothelial cells, neutrophils and macrophages is mediated by endogenously released growth factors, cytokines and chemokines, which are essential to restoring the integrity of wounded tissue [5].

Wound healing is initiated immediately after trauma by the sequential release of various growth factors and cytokines from injured blood vessels and degranulating platelets. However, if one of the phases of healing is incomplete, wound repair may be impeded, tissue recovery may be delayed, and a chronic non-healing wound may develop [6]. Localized infection may also impair healing by triggering excessive inflammation. In severe cases, this may cause the infection to become systemically disseminated [7]. Pain, especially that experienced during dressing changes, may also hinder wound healing.

In patients with chronic wounds, pain management is a major concern. Since cells release pain neuropeptides in response to a wide range of stimuli, and pain stress responses may be detrimental to healing [8,9]. These biomolecules (pain neuropeptides) also induce leukocytes and other immunoreactive cells to release proinflammatory cytokines. An imbalance of proinflammatory cytokines negatively affects wound healing, contributes to inflammatory diseases, and causes tissue destruction. Several approaches to prevent acute inflammation and pain during wound care have been evaluated. Ibuprofen-loaded materials with controlled drug release behaviors have been used to prevent excessive inflammation during the early stages of healing, and to relieve pain and promote tissue repair during later stages [7,10]. Ibuprofen is a non-steroidal anti-inflammatory drug (NSAID) used to treats acute pain, inflammation, and degenerative diseases by inhibiting cyclooxygenases (COX), which are the key enzyme in the synthesis of prostaglandins [11,12].

Ibuprofen has a short half-life [13]. It is rapidly biotransformed and must be repeatedly administered to maintain optimal plasma concentration. However, multiple ibuprofen doses may hamper treatment compliance and diminish clinical efficacy. The design of a controlled and sustained ibuprofen release system may properly address these therapeutic challenges [14,15]. Targeted therapeutic agent delivery via stimuli-responsive carriers enables on-demand drug release [16]. Likewise, localized drug delivery systems with temperature-inducible, injectable hydrogels regulate drug release and mitigate nonspecific drug distribution to healthy tissues and organs [17]. Moreover, they may help avoid excessive drug administration and improve patient compliance [18,19,20].

To enhance wound healing and attenuate bacterial colonization, various biomaterials have also been explored as bioactive dressings. These substances deliver bioactive molecules to the wound or generate materials in it with endogenous activity. Non-adhesive biomaterials that adsorb exudates and release moisture are promising for the prospect of wound dressings, and are simple to replace [21,22]. In this regard, hydrogels have attracted substantial attention due to their ability to strike an effective balance between exudate adsorption and moisture release. In addition to creating a conducive environment to initiating wound healing, they can also deliver bioactive substances to the lesion [23].

Certain hydrogels have been synthesized from naturally occurring polysaccharides due to their superior native biomimetic and physicochemical properties [24,25]. Polysaccharides are ideal biomaterials because of their excellent biocompatibility and potential reaction sites for structural modification to increase their use in biomedical applications [26,27]. Hydrogels prepared from polysaccharides alone or with other substrates [28] are characterized by high viscosity and water absorption properties [29]. Polysaccharides-based thermo-sensitive hydrogels can be obtained through different methods, such as grafting poly(ethylene glycol) (PEG) onto the chitosan backbone [30], the addition of salts into cellulose derivative (methyl cellulose) [31], and by removing ~35% of the galactose residues of xyloglucan [32]. These natural polysaccharides, including heparin, are highly recommended to ameliorate wound healing. Heparin is an anionic polysaccharide with several important biological and pharmacological activities. Heparin sulfate proteoglycans in the extracellular matrix interact with proangiogenic factors and play key roles in angiogenesis. They also protect growth factors from proteolytic degradation [33,34]. Moreover, heparin interacts with proteins and positively charged amino acid residues via its sulfate and carboxylate moieties. This interaction, either through electrostatic interaction or heparin binding domains, stabilize proteins and regulate their affinities for cell receptors. On the other hand, the selective desalfation of heparin inhibits its affinity for binding with biomolecules, and subsequently reduces its anti-coagulant effect. The fabrication of heparin-conjugated polymeric scaffolds has also been reported by several groups [35,36]. La et al. synthesized heparin-based nanospheres by conjugating heparin to amino-terminated-PLGA for wound healing applications [37]. Hence heparin-based materials have been investigated as bioactive wound dressings for accelerated wound healing or functional repairing [38,39]. Taking this into account, we prepared a N-acetylated heparin-poly(*N*-isopropylacrylamide) (Hep-PNIPAM) copolymer-based thermosensitive hydrogel by conjugating heparin to amino-terminated poly(*N*-isopropylacrylamide) (PNIPAM).

Thermoresponsive hydrogels, which undergo in situ gelation at human core body temperature, are attractive materials to be used as an alternative conventional wound dressing [40,41,42]. Several temperature-sensitive hydrogels, particularly in situ forming hydrogels, have been investigated extensively and are widely utilized as drug depots for sustained release, and can be applied with minimally invasive procedures [29,43,44]. poly-*N*(isopropylacrylamide) (PNIPAM) is one of the thermoresponsive polymers that has been explored and used as a biomaterial in various biomedical applications [45]. It undergoes a fully reversible, temperature-dependent phase transition, and has a lower critical solution temperature (LCST), ~32 °C, in aqueous solution [46,47]. However, the LCST for PNIPAM can be improved from 32 to 37 °C (body temperature) through the presence of surfactants or by conjugating with some hydrophilic copolymers [48,49,50,51]. Above the LCST, the PNIPAM polymer strongly interacts with cells and promotes cell proliferation in the hydrogel. PNIPAM-grafted copolymers have been used as scaffolds for the encapsulation of wound-healing materials and cells in cell therapy, for facilitating cell proliferation, and for tissue regeneration [52].

Here, we design a multifunctional heparin-grafted PNIPAM-based thermosensitive injectable hydrogel (Hep-PNIPAM) for the local delivery of ibuprofen and maintaining a sustained release for enhancing would healing through controlling inflammation, relieving pain and stimulating cell proliferation. The sol-gel phase transition and gelation temperatures of aqueous Hep-PNIPAM solutions were investigated, and the drug-release behaviors of ibuprofen-loaded hydrogels were characterized in vitro. The effects of the ibuprofen released from the hydrogel on proinflammatory mediators production were evaluated as indices of bacterial lipopolysaccharide (LPS)-induced cell contamination and inflammation in dermally wounded mice. The effects of ibuprofen-loaded hydrogel on tissue proliferation and wound recovery were also assessed in vivo.

## 2. Experimental Section

### 2.1. Materials

N-isopropylacrylamide, N-acetylated heparin (14 KDa), 2-aminoethanethiol hydrate, 2-(Nmorpholino) ethanesulfonic acid (MES), azobisisobutyronitrile (AIBN) and D_2_O were obtained from Sigma Aldrich (St. Louis, MO, USA). 1-Ethyl-3-[3-dimethylaminopropyl] carbodiimide hydrochloride (EDC) and N-hydroxysuccinimide (NHS) were purchased from Acros Organics (Geel, Belgium). Cellulose dialysis membrane (MWCO: 12–14 kDa) was purchased from CelluSept T1 (Braine l Alleud, Belgium). RAW264.7 cells and HaCaT cells were purchased from the BioResource collection and Research Center (Hsinchu, Taiwan). Dulbecco’s modified Eagle medium (DMEM), trypsin, penicillin and fetal bovine serum (FBS) were purchased from Gibco. Deionized water was used throughout the experiment and obtained using a Millipore water purification system. All the other chemical reagents and buffer solution components were of analytical grade.

### 2.2. PNIPAM-NH_2_ Synthesis

Amino-terminated PNIPAM (PNIPAM-NH_2_) was synthesized by free radical polymerization using AIBN as an initiator [53,54,55]. Quantities of 1.3 g (11.4 mmol) NIPAM and 8.7 mg (0.11 mmol) 2 aminoethanethiol were dissolved in 20 mL N, N-dimethylformamide (DMF). Then 24 mg (0.16 mmol) AIBN was added to the solution and the reaction mixture was heated to 75 °C for 24 h under a nitrogen atmosphere. After polymerization, the reaction mixture was concentrated by the rotary evaporation and precipitation method. Finally, it was recovered by filtration and vacuum-dried.

### 2.3. Heparin-PNIPAM Synthesis

To synthesize the PNIPAM-grafted heparin copolymer (100 mg; 0.18 mmol COOH in terms of repeating units), N-acetylated heparin was dissolved in 10 mL 0.1 M MES buffer (pH 5.6). The carboxyl group of the heparin was activated with excess EDC and NHS for 8 h at 25 °C (Scheme 1). PNIPAM-NH_2_ (1.5 g, 2 g, and 3 g per group) was then added and the reaction was performed at room temperature with stirring for 24 h. The grafting solution was purified by thermally induced precipitation at 50 °C, dialyzed (molecular weight cutoff (MWCO = 12–14 kDa) with distilled water for 4 days, and lyophilized [55,56,57].

### 2.4. Characterization of PNIPAM-NH_2_ and Hep-PNIPAM Copolymers

The chemical structure and composition of the PNIPAM and Hep-NIPAM copolymers were determined by proton nuclear magnetic resonance spectroscopy (^1^H NMR) (AVANCE 600.163 MHz; Bruker Scientific Technology Co. Ltd., Malvern, PA, USA) with deuterated chloroform (CDCl_3_) as a solvent. Fourier transform infrared (FTIR) spectra were plotted at room temperature and analyzed by attenuated total reflectance (ATR) spectroscopy (ATR-FTIR-6700; Jasco Corp., Tokyo, Japan). The number-average molecular weight (Mn) and dispersity (Đ) of PNIPAM were evaluated by gel permeation chromatography (GPC).

### 2.5. Grafting Efficiency and Ratio

The grafting efficiency (%) of Hep-PNIPAM was determined from the association between the weight of freeze-dried Hep-PNIPAM and the weights of the polymer in the feedstock (Equation (1)). The grafting ratio was also calculated using Equation (2).
(1)$$grafting\;\left(\%\right)\;=\;\frac{W_{Hep.PNIPAM\;}-\;W_{hep\;}}{W_{PNIPAM}}\;\times \;100$$
(2)$$grafting\;ratio\;=\;\frac{W_{Hep.PNIPAM}\;-\;W_{hep\;}/\;MW_{PNIPAM}}{W_{hep\;}\;/\;MW_{hep}}\;$$
where *W_hep.PNIPAM_* is the weight of freeze-dried Hep-PNIPAM and *W_hep_* and *W_PNIPAM_* are the weights of heparin and PNIPAM in the feedstock, respectively. *MW_PNIPAM_* and *MW_hep_* are the molecular weights of PNIPAM and heparin, respectively.

### 2.6. Swelling Ratio

The water content of the hydrogel was determined by a previously described method [43,44]. One milliliter of a copolymer aqueous solution was incubated in a 37 °C water bath for 1 h. The gel formed, and another 1 mL of double-distilled water was added to ensure hydrogel wettability. After 24 h of incubation at 37 °C, the supernatant was discarded and the wet hydrogel was weighed. The hydrogel swelling ratio was calculated from the weight difference of the systems before and after phase transition (Equation (3)).
(3)$$Swelling\;ratio\;=\;\frac{WH\;-\;WO}{WO}\;$$
where *WH* is the weight of the hydrated gel and *WO* is the weight of the dried test sample.

### 2.7. Hep-PNIPAM Copolymer Morphology

The copolymer hydrogel morphology was examined by field-emission scanning electron microscopy (FESEM) (JSM 6500F; JEOL Ltd., Tokyo, Japan). About 0.5 mL 10% (*w/v*) of each copolymer solution was prepared in distilled water. The copolymer solutions were placed on a silicon substrate and incubated at 37 °C until the water evaporated. The dried hydrogel on the silicon substrate was coated with platinum for 10 min and the treated samples were examined by FESEM.

### 2.8. Lower Critical Solution Temperature (LCST) and Rheological Behavior

Polymer solution turbidity or transmittance was measured at 500 nm with a UV-visible JASCO-V-650 spectrophotometer. Heparin, PNIPAM-NH2, and Hep-PNIPAM solutions (0.1 wt. %) were prepared in double-distilled water. The UV absorbances of the polymer solutions were measured from 20 to 45 °C. The polymer solution was equilibrated for 10 min at each temperature. The LCSTs of the polymer solutions were determined from the temperatures at which the solutions began to precipitate. The rheological measurements of the storage modulus (G′) and loss modulus (G″) of the hydrogels in various formulations were also carried out as functions of temperature using a Modular Compact Rheometer (MCR102). The hydrogel solution was placed on a precooled plate, and temperature sweep measurements were performed at 0.1 mm gaps between parallel plates with progressively increasing temperature with a 2 °C/min heating rate from 10 to 50 °C. The data were collected under a controlled stress (4.0 dyn/cm^2^) and a frequency of 1.0 Hz.

### 2.9. Ibuprofen Encapsulation and In Vitro Drug Release

Ibuprofen-loaded Hep-PNIPAM hydrogel (IB-Hep-PNIPAM) was prepared by the simple mixing method. Ibuprofen sodium salt (Sigma-Aldrich Corp., St. Louis, MO, USA) and 10% (*w/v*) Hep-PNIPAM were dissolved in sterile PBS. The Hep-PNIPAM solution was incubated at 4 °C to the obtained homogeneous solution. The ibuprofen was mixed with 10% (*w/v*) copolymer solution and stored at 4 °C for 10 min. The drug–copolymer solution mixture was injected into a 10 mL vial and allowed to form a gel in a 37 °C water bath. Then 2.5 mL sterile PBS (pH 7.4) was added as a release medium. At a predetermined time interval (6 h, 1–8 days), the release media were collected, and equal volumes of PBS were added. The ibuprofen content released from the gel into the medium was determined from the absorbance reading of a UV spectrometer at 264 nm.

### 2.10. In Vitro Cytotoxicity Test (MTT Assay)

To evaluate the cytotoxicity of Hep-PNIPAM on HaCaT cell lines and RAW264.7 cell lines, an MTT assay was performed as described in previous studies [20,58,59,60]. Briefly, HaCaT cells and RAW264.7 cells were seeded in a 96-well plate at a density of 1 × 10^4^ cells/well. The cells were cultured in DMEM containing 10% (*v/v*) FBS, 1% (*w/v*) sodium pyruvate, and 1% (*w/v*) streptomycin at 37 °C in a humidified atmosphere with 5% CO_2_. After 24 h, the culture medium was removed and the cells were washed twice with PBS. Serial dilutions of Hep-PNIPAM (control (0), 1, 0.5, 0.1, 0.02, and 0.01 mgmL^−1^) were prepared with fresh media, and added to the cells, which were then incubated for another 24 h. The cells were then washed with PBS and 20 μL MTT solution (5 mgmL^−1^ in PBS) was added to each well along with 100 μL culture medium. The cells were incubated for an additional 4 h, after which the MTT solution was discarded and 100 μL dimethyl sulfoxide (DMSO) was added to solubilize the formazan crystals and measure its absorbance at 570 nm via an enzyme-linked immunosorbent assay (ELISA). The viability of the RAW264.7 cells was determined after subjecting them to varying concentrations of IB-Hep-PNIPAM in the presence or absence of 1 μgmL^−1^ LPS for 24 h. Absorbance at 570 nm was measured via a microplate reader (ThermoMultiskan FC microplate photometer; Thermo Fisher Scientific, Waltham, MA, USA). Cell viability (%) was calculated according to Equation (4).
(4)$$Cell\;viability\;\left(\%\right)\;=\;\frac{\mathrm{Absorbance}\;\mathrm{of}\;\mathrm{test}}{\mathrm{Absorbance}\;\mathrm{of}\;\mathrm{control}}\;\times \;100\;$$

HaCaT cell viability was also assessed using a live/dead assay. The cells were seeded in a confocal dish (1 × 10^5^ cells/dish) and cultured in DMEM containing 10% (*v/v*) FBS, 1% (*w/v*) sodium pyruvate, and 1% (*w/v*) streptomycin. The cells were treated with 1 mg/mL Hep-PNIPAM and incubated at 37 °C in a humidified atmosphere with 5% CO_2_. After 24 h incubation, the medium was removed and calcein acetoxymethyl ester (calcein AM; 2 μM) and propidium iodide (PI; 4 μM) were added to the cell. The cells were incubated for an additional 30 min at 37 °C with 5% CO_2_, washed thrice with PBS, and imaged using a confocal microscope.

### 2.11. Anti-Inflammation Assay for Ibuprofen-Loaded Hydrogel

#### 2.11.1. Nitric Oxide (NO) Level Measurement

RAW264.7 cells were seeded in a 24-well plate at 1x10^5^ cells/well and treated with a supernatant of culture medium taken after 10 wt. % Hep-PNIPAM loaded with ibuprofen (IB-Hep-PNIPAM) was incubated. After 1 h of treatment, 1 μgmL^−1^ LPS was added and the cells were incubated for an additional 24 h at 37 °C with 5% CO_2_. The anti-inflammatory effects of IB-Hep-PNIPAM and free IB were also compared by treating cells with 20 μM free IB and equivalent concentrations of IB-loaded Hep-PNIPAM. Then 100 μL of cell culture supernatant was mixed with 100 μL Griess reagent (1% (*w/v*) sulfanilamide in 5% (*v/v*) phosphoric acid and 0.1% *N*-(1-naphthyl)ethylenediamine dihydrochloride) and incubated at room temperature for 15 min. The NO concentration was determined at 540 nm ((ELISA) reader (ThermoMultiskan FC microplate photometer; Thermo Fisher Scientific, Waltham, MA, USA) and interpolated from a sodium nitrite (NaNO_2_) standard curve.

#### 2.11.2. TNF-α and PGE_2_ Assays

RAW264.7 cells were seeded in 24-well plates and subjected to 10 wt. % Hep-PNIPAM loaded with serial dilutions of ibuprofen (IB-Hep-PNIPAM), 20 μM IB-loaded Hep-PNIPAM, and 20 μM free IB for 1 h followed by incubation with 1 μgmL^−1^ LPS for 24 h at 37 °C under 5% CO_2_. The PGE2 and TNF-α levels in the culture medium were determined by ELISA (Thermo Fisher Scientific, Taipei City, Taiwan) according to the manufacturer’s instructions.

### 2.12. Animal Experiment

Eight-week old male BALB/c mice, each weighing 24–26 g, were purchased from BioLASCO Taiwan Co., Ltd., Taipei City, Taiwan. The animals were maintained under a light–dark cycle (12 h–12 h) and had free access to commercial mouse chow (Nutrition International LLC, Brentwood, MO, USA) and distilled water. All animal care and in vivo trials were approved by the Research Ethics and Animal Welfare Committee and the experimental procedures were performed according to the guidelines of the Institutional Animal Care and Use Committee (IACUC-16-168) in the National Defense Medical Center (accredited by the Association for Assessment and Accreditation of Laboratory Animal Care International).

#### 2.12.1. Wound Healing Rate

Mice were divided into the following 3 groups (n = 5 per group): (1) PBS-treated wound (control), (2) Hep-PNIPAM-treated wounds, and (3) IB-Hep-PNIPAM-treated wounds. The mice were anesthetized by intramuscular Zoletil (20 mgkg^−1^) injection [20]. Their dorsal skin was shaved with an Oster Mark II animal clipper before creating the wounds. Full-thickness skin wounds, 6 mm in diameter, were induced on the dorsal skin with a biopsy punch after skin disinfection with 70% (*v/v*) ethanol. Hep-PNIMAM (10% (*w/v*)) and IB-Hep-PNIPAM (10% (*w/v*) (Hep-PNIMAM loaded with 10 mgmL^−1^ equivalent ibuprofen) were topically applied to the wound areas. The wound areas were measured by tracing the wound margins with a Vernier caliper. Changes over time were expressed as % of the initial wound areas (Equation (5)) [61].s
(5)$$Wound\;area\;=\;\frac{Area\;\left(day\;N\right)}{Area\;\left(day\;0\right)}\;\times \;100\;$$
where Area (day 0) is the initial wound area at day 0 and Area (day N) is the area on day N after wounding. Mice were sacrificed at 15 days post-wounding. The skin and major organs (lung, heart, liver, kidney and spleen) were excised and fixed in 3.7% (*v/v*) paraformaldehyde. Hematoxylin and eosin (H&E) staining was conducted per the manufacturer’s instructions and previously reported methods.

#### 2.12.2. In Vivo Anti-Inflammation Assay for Ibuprofen-Loaded Hydrogel

For the in vivo anti-inflammation study, the mice were randomly segregated into the PBS, LPS (10 mgkg^−1^), and ibuprofen-loaded Hep-PNIPAM (IB-Hep-PNIPAM) treatment groups (n = 5 per group). The mice were wounded as previously described and subjected to the aforementioned treatments administrated intradermally at the wound area. After 24 h, the IB-Hep-PNIPAM group was induced with LPS (10 mgkg^−1^) through intradermal injection. Then 12 h after LPS injection, the mice were sacrificed, and their blood was collected and centrifuged at 2000× *g* for 20 min to obtain the serum. Serum PGE_2_ expression was determined by ELISA (Thermo Fisher Scientific, Taipei City, Taiwan) and NO secretion was measured with a nitrite assay kit (Griess reagent) per the manufacturer’s instructions. H&E staining was conducted per the manufacturer’s instructions and previously reported methods.

### 2.13. Statistical Analysis

All data were recorded as the mean ± the standard deviation. Statistical analysis of the results was performed using one-way ANOVA and the least significant difference test. A p value <0.05 was statistically considered to be significant, and *p* < 0.01 was considered highly significant. All experiments were performed at least three times.

## 3. Results and Discussion

### 3.1. Synthesis of Hep-PNIPAM Copolymer

The chemical structures of heparin, PNIPAM-NH_2_, and the copolymer were predicted from the FTIR spectra. As shown in Figure 1, the heparin spectrum (black) showed peaks at 2979 cm^−1^ (−**CH** stretch), 1739 cm^−1^ (−**COO**−) and 1216 cm^−1^ (−**CO** stretch). These values confirmed the basic heparin structure. The amino-terminated PNIPAM-NH_2_ spectrum (red) showed functional group peaks at 2979 cm^−1^ and at 1548 cm^−1^ attributable to (−**CH_3_**) stretching and the PNIPAM-NH_2_ amide II bonds, respectively. The peak at 3251 cm^−1^ for PNIPAM-NH_2_ represented the substituted amine functional group’s stretching vibration. The functional peak of Hep-PNIPAM (−**COO**−) stretching at ~1739 cm^−1^ was lower than that of heparin. Thus, PNIPAM-NH2 was coupled to heparin [51,62]. The ^1^H NMR for the copolymer (Figure 2) showed characteristic PNIPAM peaks at **a** (δ = 1.20 ppm), **d** (δ = 1.7 ppm), and **b** (δ = 4.00 ppm), and for heparin at **H** (δ = 2.1 ppm, 3.5 ppm, and 4.4 ppm). The FTIR spectrum and the ^1^H NMR proved that Hep-PNIPAM was successfully synthesized. The number-average molecular weight (Mn) and the dispersity (Đ) for PNIPAM-NH_2_ were also evaluated by gel permeation chromatography (GPC). According to the result, the molecular weight was 11,230 g/mol with a narrow dispersity (Đ) (1.106), which further confirmed the polymerization of the copolymer (Appendix A
Appendix A).

### 3.2. Grafting Efficiency and Swelling Ratios

The molar ratio of PNIPAM-NH_2_ to heparin during Hep-PNIPAM synthesis was determined from the number of PNIPAM chains grafted onto a heparin backbone. By controlling the molar ratio of PNIPAM-NH_2_ to heparin during grafting, three batches of copolymers with different grafting ratios were obtained. As shown in Table 1, the grafting ratios of Hep-PNIPAM-b_1_, Hep-PNIPAM-b_2_, and Hep-PNIPAM-b_3_ were 10, 17, and 23, respectively. The conjugation yields were 70%, 96%, and 84% for Hep-PNIPAM-b_1_, Hep- PNIPAM-b_2_, and Hep-PNIPAM-b_3_, respectively. The swelling ratio of Hep-PNIPAM was evaluated according to the weight differences of the systems before and after phase transition. The swelling ratios of PNIPAM-NH_2_, Hep-PNIPAM-b_1_, Hep-PNIPAM-b_2_, and Hep-PNIPAM-b_3_ were 5, 22, 14, and 12, respectively. Hydrogel morphology was examined by FESEM. Appendix A shows that the hydrogels had a microporous inner structure. Their pore sizes increased with heparin concentration (ranges about 0.5 µm to 3 µm in diameter). The pores in the hydrogel could encapsulate ibuprofen for sustained release.

### 3.3. Lower Critical Solution Temperature (LCST) and Rheological Behavior

Figure 3a shows the temperature-mediated transmittances for heparin, PNIPAM-NH_2_, and copolymer. The solution temperatures were maintained for ~10 min per measurement [55,63,64]. Heparin transmittance was steady at ~98.0%, whereas for the PNIPAm-NH_2_ solution it decreased from 99% to 2.0% as the temperature rose to 32 °C. The transmittances of the copolymers (Hep-PNIPAM-b_1_, Hep-PNIPAM-b_2_, and Hep-PNIPAM-b_3_) declined from 99% to 21%, 12%, and 3%, respectively, at about body temperature. The confirmed LCST of PNIPAM-NH_2_ was ~28 °C and the predicted LCSTs of the copolymers (Hep-PNIPAM-b_1_, Hep-PNIPAM-b_2_, and Hep-PNIPAM-b_3_) were ~34 °C, ~32 °C, and ~30.5 °C, respectively. LCST modification to high temperatures for copolymers with different PNIPAM grafting ratios on heparin may be influenced by the grafting ratio on the transmittance [51,55]. The copolymers with relatively higher grafting ratios may be explained by the increased hydrophobicity, in which the LCST of the copolymers shifted to that of PNIPAM-NH_2_. In contrast, the copolymers with comparatively lower grafting ratios (Hep-PNIPAM-b_1_) had a shorter hydrophobic segment and a higher LCST value. Hydrogel rheology was measured as a function of temperature. The storage modulus (G′) and loss modulus (G″) reflect the elasticity or energy stored in the material during deformation, and its viscosity or energy consumed in the form of heat, respectively [65]. The hydrogel forms when G′ > G″. In contrast, when the polymeric solution remains in the liquid state, G′ < G” [66,67]. The gel point is located at the intersection of G′ and G″. Figure 3b shows the changes in G′ and G” for 10 wt. % PNIPAM-NH_2_, Hep-PNIPAM-b_1_, Hep-PNIPAM-b_2_, and Hep-PNIPAM-b_3_ as functions of temperature. The gel points of PNIPAm-NH_2_, Hep-PNIPAM-b_1_, Hep-PNIPAM-b_2_ and Hep-PNIPAM-b_3_ were ~33.1 °C, ~36.9 °C, ~35.2 °C and ~34 °C, respectively. The copolymer gelation temperature increased with the decreasing grafting ratio. PNIPAM-NH_2_ and the copolymers could undergo in situ hydrogel formation at human core body temperature. From these three batches, the higher gelation temperature (approach to body temperature) of Hep-PNIPAM-b_1_ was selected for further investigation (in vitro cell viability and in vivo studies).

### 3.4. Ibuprofen Encapsulation and In Vitro Drug Release

Several studies reported that ibuprofen powerfully modulates wound healing [7,10]. Ibuprofen is a commonly used NSAID to treat acute pain, inflammation, and degenerative diseases. Ibuprofen was encapsulated in the Hep-PNIPAM hydrogel by simple mixing. In vitro drug release behavior in phosphate-buffered saline PBS (pH 7.4) was explored. Ibuprofen was released mainly by diffusion and was followed by hydrogel erosion. The ibuprofen release profiles from the Hep-PNIPAM hydrogels are shown in Figure 4b. The amount of ibuprofen released was measured at 264 nm by UV spectrophotometry using the interpolation of a free ibuprofen calibration curve (Figure 4a) [68]. As ibuprofen is an NSAID, its efficacy is vital during the inflammatory phase of healing. The drug release curve revealed that sustained ibuprofen release lasted for 1 week, and the cumulative release was ~70%. The drug release was rapid within the first 8 h (~15%), which indicated a light burst release pattern that is effective in acute inflammation and pain treatment.

### 3.5. In Vitro Cytotoxicity Test (MTT Assay)

To quantify cell proliferation and viability, the MTT assay is accurate and simple. In vitro copolymer biocompatibility with human immortalized keratinocyte HaCaT cell lines was assessed by MTT assay [60]. Cell viability was estimated after 24 h incubation with varying concentrations of Hep-PNIPAM copolymer (0.01–1 mg mL^−1^), and the result showed that the copolymer has a negligeable toxic effect against HaCaT cells (Figure 5a). The MTT assay on LPS-induced RAW264.7 cells also indicated that IB-Hep-PNIPAM was not toxic up to ibuprofen concentrations of 201 μM (Figure 5b). These results verify that both the Hep-PNIPAM and ibuprofen-loaded copolymers were not toxic to normal cells. The cytocompatibility of Hep-PNIPAM with HaCaT cells was also confirmed by cell fluorescence imaging (see Appendix A). The green fluorescence indicates the live cells, while the red fluorescence indicates the dead cells.

### 3.6. Effects of IB-Hep-PNIPAM on NO, PGE_2_, and TNF-α Production in LPS-Stimulated RAW264.7 Cells

Previous studies have reported that ibuprofen has been loaded into the hydrogel to enhance anti-inflammatory activity and wound healing [69]. This study focuses on reducing inflammatory reaction and accelerating wound healing through thermosensitive IB-loaded Hep-PNIPAM hydrogel. To assess the anti-inflammatory effects of ibuprofen-loaded Hep-PNIPAM hydrogel, the NO, PGE_2_, and TNF-α levels were measured in RAW264.7 macrophages [70,71,72]. Cells were pretreated with ibuprofen-loaded Hep-PNIPAM hydrogel for 1 h and stimulated with LPS for 24 h. The LPS stimulation of IB-Hep-PNIPAM-treated RAW264.7 cells significantly upregulated the proinflammatory mediators (NO, PGE_2_, and TNF-α) as compared with cells which were not stimulated with LPS. However, the pretreatment of RAW264.7 cells with IB-Hep-PNIPAM decreased the production of these proinflammatory mediators in RAW264.7 cells stimulated with LPS in a dose-dependent manner (Figure 6a,c,d). The production of each proinflammatory mediator was calculated according to the calibration curves (Appendix A). The suppression of proinflammatory mediator secretion by IB-Hep-PNIPAM was comparable to that of free ibuprofen, and was significantly downregulated as compared to LPS-induced cells (Figure 6b,e). Thus, IB-Hep-PNIPAM inhibits LPS-induced inflammation by suppressing the secretion of the proinflammatory mediators NO, PGE_2_, and TNF-α in RAW264.7 macrophages. The results are consistent with the previous studies [73].

### 3.7. In Vivo Wound Healing Effects and Healing Rate

As both hydrogel and ibuprofen have been proven to be beneficial in wound healing, it is expected that the combination of hydrogel and ibuprofen would be even more efficacious in wound healing. Hep-PNIPAM-b1 polymer was selected for use in the in vivo study, as it had the highest swelling ratio and formed a hydrogel close to normal body temperature (37 °C). Wounds on the backs of mice were induced with a biopsy punch. Figure 7a shows the appearance of the wounds treated either with hydrogel or PBS. Wound healing was evaluated at 0 d, 3 d, 5 d, 7 d, 9 d, 11 d, 13 d, and 15 d. Figure 7b shows that all wound areas decreased with time. The wounds treated with IB-Hep-PNIPAM and Hep-PNIPAM had higher closure ratios than the control group. Since the mice studied for wound closure had no inflammatory challenge, there was no significant difference between Hep-PNIMAM and IB-Hep-PNIPAM in terms of the effect on wound healing. However, from the result we realized that if the mice model was diabetic or cancer-induced with excessive inflammation, the effect of IB-loaded Hep-PNIPAM would be more noticeable than Hep-PNIPAM. In a previous study, Van-Linh Nguyen et al. [61] confirmed that anti-inflammatory drugs could accelerate infected wounds due to their antibiotic properties. In this study, an anti-inflammatory drugs-loaded hydrogel was used to enhance wound healing in vivo on a mouse model. The drug released from the hydrogel had a role in reducing inflammation. It was investigated both in vivo and in vitro with LPS-induced cells and mice, respectively. New epidermis grew from the margin to the center of the wound bed and decreased the wound depth and area. Histological analysis showed that granulation and epithelization proceeded faster in the IB-Hep-PNIPAM than the PBS-treated wounds (Figure 7c). Other organ histology examinations confirmed that the hydrogel was nontoxic, and there were no signs of inflammation or infection in hydrogel-covered wounds. Hydrogels combined with analgesics create a moist environment that prevents dehydration, absorb exudate, alleviate pain, and accelerate wound healing. This was demonstrated by the ibuprofen-loaded hydrogel (IB-Hep-PNIPAM), which accelerated wound healing.

### 3.8. In Vivo Anti-Inflammation Assay on Ibuprofen-Loaded Hydrogel

Endometrial epithelial and immune cell stimulation with pathogens may induce cytokinesecretion [74]. Mice stimulated with LPS presented with significant increases in TNF-α and NO secretion see (Figure 8). However, pretreatment with IB-Hep-PNIPAM downregulated LPS-induced NO (Figure 8a) and TNF-α (Figure 8b) production by ~50% and ~35%, respectively, compared with LPS alone. To examine histological changes in response to treatment, tissue slides were made from skin, liver, and heart tissue samples. Inflammatory cell infiltration was observed in the skin, livers, and hearts of mice subjected to LPS (Figure 8c). However, no such infiltration was observed in the same tissues taken from mice treated with IB-Hep-PNIPAM or the control (PBS). Therefore, IB-Hep-PNIPAM is a potential anti-inflammatory agent.

## 4. Conclusions

A thermosensitive PNIPAM-conjugated heparin hydrogel was prepared as a wound healing material and loaded with ibuprofen. It was formulated as an injectable solution for enhanced healing efficacy. Ibuprofen was released from the Hep-PNIPAM hydrogel in a sustained manner and inhibited the LPS-induced activation of the proinflammatory mediators NO, TNF-α, and PGE2. Hydrogels create a moist environment on wounds, absorb exudates, and prevent dehydration. In combination with anti-pain drugs, they create an environment conducive to wound healing. In the mouse wound model, IB-Hep-PNIPAM hydrogels promoted rapid healing and were anti-inflammatory. The combination of ibuprofen and Hep-PNIPAM injectable hydrogel improved wound healing, and is a promising candidate for this type of therapy.

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
