# Peer review of "Ibuprofen-Loaded Heparin Modified Thermosensitive Hydrogel for Inhibiting Excessive Inflammation and Promoting Wound Healing"

_polymers, 2020, doi:10.3390/polym12112619_

Round 1

Reviewer 1 Report

In this manuscript, the authors report on the preparation of thermoresponsive (injectable) hydrogels based on heparin-g-PNIPAM copolymers able to be loaded with ibuprofen, as well their evaluation in vitro (cytotoxicity, inhibition of inflammation) and in vivo in mice (wound healing). The approach of combination of PNIPAM (thermogelling property) and heparin (reported beneficial compound for wound healing) is interesting, and has not been reported so far in the context of wound healing. The manuscript is clear and, overall, the work is well conducted. There are however some issues that should be addressed:

1) The characterization of copolymers should be strengthened.

-The authors performed a dialysis against a membrane of 12-14 kDa cutoff. Are they sure that PNIPAM not coupled is entirely removed during dialysis step? Indeed, Mn of PNIPAM is 11230 g/mol, very close to this cutoff (and dispersity is not provided, but considering the distribution, it is highly probable that some chains are higher in molecular weight than this cutoff value). Thus, remaining free PNIPAM homopolymer could induce errors in grating ratios, which are only determined by weighting. Also, the molecular weight of heparin is not given.

- Despite NMR/FTIR analysis (which shows presence of both heparin and PNIPAM in the final product), the covalent character of the coupling of PNIPAM-NH2 on heparin is not really proved. This could be for example performed by quantifying the decrease in amine functions following amide formation using standard methods (TNBS, fluorescamine…), and would give access to grafting ratio.

2) P. 12, line 397-398: “In contrast, Hep-PNIPAM (without ibuprofen) did not inhibit proinflammatory mediator production”. This is not so clear from the experiments (Figure 6), since the samples “cells/LPS” and “cells/LPS/hydrogel alone” are not compared.

3) Other minor points:

  • p4: molecular weight of N-acetylated heparin is not given.
  • p4: it is claimed that 2-aminoethanedithiol is used as a chain transfer agent in the synthesis of PNIPAM-NH2, but it does not fit with the structure given in Scheme 1 (propane based agent, i.e. 3-aminopropane-1-thiol). Please clarify this point.
  • p6, line 169: references 43 and 44 seem to be inappropriate.
  • p6, line 189: word not correct:“meausyrment”.
  • The “ELISA” term seems to be sometimes wrongly used (p.7, line 216-217, line 239,…).
  • p8, line 295-296: this FTIR assignment (1650 cm-1) seems wrong (not amine but rather C=O of the amide bonds In PNIPAM).
  • p8: PDI of PNIPAM is not given in Figure S1. In addition, the term “dispersity” (Đ) should be now adopted instead of “polydispersity index” (PDI).
  • p10, line 318-319: Average pore size values should be given in the text. In figure S2, the scale bars are not readable. Please make them readable.
  • p10, line 326: I guess the authors meant “32°C” instead of “42°C”.
  • p11, Figure 3(a): there is a typo in the legend: “PEG-PNIPAM-3”.
  • p12, line 375: “the copolymers presented with no cytotoxicity”; P. 14, line 418: “wounds treated ... presented with higher closure…” (with?).
  • p15, line 449: replace “Figure S5” by “Figure 8c”.

Author Response

Authors’ responses to reviewers’ comments

Reviewer-#1

The approach of combination of PNIPAM (thermogelling property) and heparin (reported beneficial compound for wound healing) is interesting and has not been reported so far in the context of wound healing. The manuscript is clear and, overall, the work is well conducted. There are however some issues that should be addressed:

  1. The characterization of copolymers should be strengthened. The authors performed a dialysis against a membrane of 12-14 kDa cutoff. Are they sure that PNIPAM not coupled is entirely removed during dialysis step? Indeed, Mn of PNIPAM is 11230 g/mol, very close to this cutoff (and dispersity is not provided, but considering the distribution, it is highly probable that some chains are higher in molecular weight than this cutoff value). Thus, remaining free PNIPAM homopolymer could induce errors in grating ratios, which are only determined by weighting. Also, the molecular weight of heparin is not given.

Despite NMR/FTIR analysis (which shows presence of both heparin and PNIPAM in the final product), the covalent character of the coupling of PNIPAM-NH2 on heparin is not really proved. This could be for example performed by quantifying the decrease in amine functions following amide formation using standard methods (TNBS, fluorescamine…), and would give access to grafting ratio.

Responses #1: The authors appreciated the reviewer’s concern and comments. All the points raised by the reviewer was addressed accordingly as follow:    

The dialysis process employed in our study (using a membrane cut off 12-14 kDa) is sufficient to remove unreacted PNIPAM. In the first step, the pore size of the membrane is wider enough to allow the outward flux of molecules with molecular weight below 12,000 g/mol (Mn of PNIPAM-NH2 was 11230 g/mol, and 14000 g/mol for Heparin). To further enhance the removal of unreacted PNIPAM-NH2, we exchange the DI water in 4 h intervals for extended time (4 days), already mentioned in the manuscript. Inevitably, the dispersity of the resulting Hep-PNIPAM would be relatively higher as the reaction involved grafting of the heparin backbone with PNIPAM-NH2. In such reactions, little variation in the number of pendant groups (even a single PNIPAM-NH2 group) culminated in higher dispersity as compared to ordinary mainchain products. Thus, we believe that the purification method we followed allowed to collect sufficiently pure Hep-PNIPAM conjugate for further characterization. The molecular weight of heparin was also mentioned in the revised manuscript which is14 kDa (Section 2.1).       

Regarding to characterization of heparin grafted PNIPAM, we already characterized the conjugation of heparin with PNIPAM using 1H NMR, FTIR and GPC. In order to quantify the amine functions using the aforementioned techniques you suggested, the heparin backbone should contain reactive -NH2 groups, but the heparin we used contain repeating -COOH group which can bind with the amine functional group of PNIPAM to form amide bond. Thus, it is impossible to compare the alterations in the number of amine groups before and after coupling with PNIPAM (as the -NH2 group belonged to the grafting agent rather than the heparin chain).

2. 12, line 397-398: “In contrast, Hep-PNIPAM (without ibuprofen) did not inhibit     proinflammatory mediator production”. This is not so clear from the           experiments  (Figure 6), since the samples “cells/LPS” and “cells/LPS/hydrogel   alone” are not compared.

Response #2: We acknowledged the issues you pointed out and to avoid ambiguity of the readers, we have removed the sentence which was not relevant for our general conclusion of the results.  

3. Other minor points:

  1. p4: molecular weight of N-acetylated heparin is not given.
  2. p4: it is claimed that 2-aminoethanedithiol is used as a chain transfer agent in the synthesis of PNIPAM-NH2, but it does not fit with the structure given in Scheme 1 (propane based agent, i.e. 3-aminopropane-1-thiol). Please clarify this point.
  3. p6, line 169: references 43 and 44 seem to be inappropriate.
  4. p6, line 189: word not correct:“meausyrment”.
  5. The “ELISA” term seems to be sometimes wrongly used (p.7, line 216-217, line 239).
  6. p8, line 295-296: this FTIR assignment (1650 cm-1) seems wrong (not amine but rather C=O of the amide bonds In PNIPAM).
  7. p8: PDI of PNIPAM is not given in Figure S1. In addition, the term “dispersity” (Đ) should be now adopted instead of “polydispersity index” (PDI).
  8. p10, line 318-319: Average pore size values should be given in the text. In figure S2, the scale bars are not readable. Please make them readable.
  9. p10, line 326: I guess the authors meant “32°C” instead of “42°C”.
  10. p11, Figure 3(a): there is a typo in the legend: “PEG-PNIPAM-3”.
  11. p12, line 375: “the copolymers presented with no cytotoxicity”; P. 14, line 418: “wounds treated ... presented with higher closure…” (with?).
  12. p15, line 449: replace “Figure S5” by “Figure 8c”.

Responses for #i-xii:

We appreciate very much for the valuable comments and have thoroughly revised our manuscript to accommodate all the comments and issues raised point-by-point.

Responses- #i: molecular weight of N-acetylated heparin was mentioned in the revised manuscript as 14 kDa

Responses- #ii: The structure of 2-aminoethanedithiol in the scheme was corrected

Responses -#iii: The references were replaced by appropriate one in the revised manuscript

  1. Bubpamala, T., K. Viravaidya-Pasuwat, and P. Pholpabu, Injectable poly (ethylene glycol) hydrogels cross-linked by metal-phenolic complex and albumin for controlled drug release, ACS Omega, 5(31) (2020) 19437-19445.
  2. Liu, Y., Xia, M., Wu, L., Pan, S. Zhang, Y., He, B., He, P., Physically cross-linked double-network hydrogel for high-performance oil–water separation mesh. Ind. Eng. Chem. Res., 58(47) (2019) 21649-21658.

Responses- #iv: The word: “meausyrment” was corrected and replaced by “measurement”.

Responses- #v: Wrongly mentionedELISA” was changed by microplate reader in the revised manuscript

Responses- #vi: We admitted the reviewer’s comment, and it was corrected in the revised manuscript that the new spectrum at 3251 cm-1 was represented the substituted amine group.

Responses- #vii: The term “dispersity” is adopted throughout the manuscript and the dispersity (Đ) of PNIPAM-NH2 was given in Figure S1 of the revised manuscript

Responses- #viii: The size of the pore was measured and the scale bar for figure S2 was amended in the supporting information as well as in the revised manuscript.

Responses- #ix: Yes, it was corrected in the revised manuscript, “32 °C”

Responses- #x: it was corrected as “Hep-PNIPAM-3”

Responses- #xi:  The sentence was modified in the revised manuscript as “ Cell viability was estimated after 24 h incubation with varying concentrations of Hep-PNIPAM copolymer (0.01–1 mg mL-1) and the result showed that the copolymer has negligeable toxic effect against HaCaT cells (Figure 5a), and … The wounds treated with IB-Hep-PNIPAM and Hep-PNIPAM had higher closure ratios than the control group”.

Response #xii: “Figure S5” in the manuscript was corrected as “Figure 8c”

Reviewer 2 Report

Manuscript entitled "Ibuprofen-loaded heparin modified thermosensitive hydrogel for inhibiting excessive inflammation and promoting wound healing" by authors Abegaz Tizazu Andrgie et al., is well written. Authors have made an attempt to prepare hydrogel formulation with ibuprofen (non-selective COX inhibitor) and evaluated the wound healing activity by in vivo method and anti-inflammatory activity by in vitro and in vivo methods. Authors have also evaluated the formulation and determined the cytotoxicity of the formulation using cell culture studies. 

Kindly address the following minor comments

  1. Line # 189- minor typo error (measurement)
  2. Line # 252 - Please confirm, is it distilled water or de-ionized water
  3. Line # 391, 392 is not clear. Please clarify, Do LPS treated or untreated cells show upregulated proinflammatory mediators
  4. Fig. 5 & 6: please provide the rationale for selecting 2, 10, 20, 41, 51, 101 & 201 uM concentrations
  5. Please mention the route of LPS administration to mice
  6. Since the formulation forms hydrogel at 37 degree temperature, how was the cell culture studies performed, How was the LPS challenge done with cells post treatment with formulation?
  7. Was the ibuprofen quantified in serum/blood 
  8. Apart from wound size and hitopath, any biochemical parameters studies at wound site?
  9. Supplemental results were not discussed in the manuscript
  10. How was the formulations were applied to the wound? Kindly elaborate in the methods section. 

Author Response

Authors’ responses to reviewers’ comments

Reviewer - #2

The Authors are kindly asked to address the following minor comments

  1. Line # 189- minor typo error (measurement)

Responses- #1: The word: “meausyrment” was corrected and replaced by “measurement”.

  1. Line # 252 - Please confirm, is it distilled water or de-ionized water

Responses - #2: We used distilled water for drink by laboratory mice.

  1. Line # 391, 392 is not clear. Please clarify, Do LPS treated or untreated cells show upregulated proinflammatory mediators

Response -#3: The sentences were amended and clarified in the revised manuscript as follow: “LPS stimulation of IB-Hep-PNIPAM treated RAW264.7 cells were significantly upregulated the proinflammatory mediators (NO, PGE2, and TNF-α) as compared with cells which were not stimulated with LPS. However, the pretreatment of RAW264.7 cells with IB-Hep-PNIPAM decreased production of these proinflammatory mediators in  RAW264.7 cells stimulated with LPS in a dose-dependent manner (Figure 6a, c &d)”.

  1. 5 & 6: please provide the rationale for selecting 2, 10, 20, 41, 51, 101 & 201 uM concentrations

Responses -#4: We just randomly select serial concentration to study the dose dependent toxic effects of IB on cells by considering the maximum dose we would use for its anti-inflammatory activity.

  1. Please mention the route of LPS administration to mice

Responses - #5: LPS was administered intradermally at the wound area, it was mentioned in the revised manuscript (Section 2.12.2)

  1. Since the formulation forms hydrogel at 37°C temperature, how was the cell culture studies performed, How was the LPS challenge done with cells post treatment with formulation?

Responses -#6: Inorder to study the anti-inflamatory effect of drug-loaded hydrogel (IB-Hep-PNIPAM) againes RAW264.7 cells in vitro, first the drug-loaded hydrogel was incubated in the culture medium and then the medium was taken to be used for culturing the cells in a 24-well plate. We did not applied the hydrogel directlly to the cells. Following this (after 1 h treatment), the cells were treated with LPS (1 μgmL-1 ) for 24 h (discribed brifly in Section 2.11.1 of the manuscript)

  1. Was the ibuprofen quantified in serum/blood?

Responses -#7: Alghough ibuprofen in the blood/serum was not quantified in this study, the amount of IB loaded in the hydrogel and its release rate was known. Thus, an equivalent amount of 10 mg/kg of IB was applied to each mouse in the form of IB-Hep-PNIPAM hydrogel and according to the in vitro release behavior, nearly 70% of its content was released within 5 days. Therefore, we assumed that the amount of IB in the blood can be estimated indirectly as per its release rate.

  1. Apart from wound size and hitopath, any biochemical parameters studies at wound site?

Responses -#8: In this study our interest was to investigate an extended anti-inflammatory effect of IB while it was applied locally using thermosensitive Hep-PNIPAM hydrogel and its subsequent enhanced wound healing effects. As a result, we only studied the level of production of proinflammatory molecules (both in vitro and in vivo), pathohistology of major organs and wounded tissue, and the rate of wound healing (would size) on wounded mice model.

  1. Supplemental results were not discussed in the manuscript

Response -#9: The data found in the supplementary material was discused in the rivised manuscript.

  • Molecular weight of PNIPAM-NH2 copolymer (Fig. S1) was discussed in section 3.1, page 18 of the manuscript.
  • Porous structure of the hydrogel and its size (Fig. S2) was described in section 3.2, page 20 of the manuscript.
  • Cell viability test using Calcein AM/PI (Fig. S3) was also discussed in section 3.5, page 24 of the revised manuscript and
  • The calibration curve (Fig. S4) was referred in page 25 of the revised manuscript.
  1. How was the formulations were applied to the wound? Kindly elaborate in the methods section. 

Responses -#10: The way how to apply different formulations of treatments (Saline (control), Hep-PNIPAM, and IB-Hep-PNIPAM) was briefly explained in the method section (Section 2.11.2) of the revised manuscript.

Round 2

Reviewer 1 Report

The authors have correctly addressed the concerns raised in my first review and the paper can now be accepted for publication (regarding grafting of PNIPAM-NH2 on heparin, I meant to perform amine quantification, before and after coupling, in the crude reaction mixture, not on the purified product...).

This manuscript is a resubmission of an earlier submission. The following is a list of the peer review reports and author responses from that submission.

Round 1

Reviewer 1 Report

The authors developed an ibuprofen-encasulated hydrogel and tested its efficacy in skin wound healing, in mice, evaluating its anti-inflammatory activity in vitro and in vivo. I have some points to discuss.

Were the mice treated everyday? I did not find this information.

Did the rats received any drug to minimize post-operatory pain? If they did not, why not?

N=5 mice is a very low number to wound healing assay. Each mouse has its own response, and we use at least 7 animals per group. The same for ELISA assay.

"All experiments were performed at least three times." Did the authors perform the in vivo assay three times? Using n=5 three times??

"the mice were sacrificed (using CO2)" Why did authors use this controversial way to euthanize (not "sacrifice")? Was this approved by the ethics committee? 

Page 12, line 397: "This was demonstrated by the ibuprofen-loaded hydrogel (IB-Hep-PNIPAM), which accelerated wound healing". I can not see this result in the figure 7b. Clearly, HEP-PNIPAM and IB-Hep-Pnipam present the same healing efficacy. Is there any statistical difference any day? Did you evaluate the healing progress or only after 15 days? Other point is, at the last day, the three lines in the graph are in the same point, indicating that hep, ib-hep and control equally healed the wounds.

Discard the organs histology. We can see nothing in the photomicrographies (not "histological observations"). "Other organ histology examinations confirmed that the hydrogel was nontoxic and there were no signs of inflammation or infection in hydrogel-covered wounds." It is not correct to conclude that the hydrogel was nontoxic using these photomicrographies. At least, 40x objective lens is needed in this case.

The same for liver and heart in 8c, they are out of focus. What is the explanation for inflammatory infiltrate in liver and heart? It is not in the text.

Reviewer 2 Report

1. General remarks 

  • The manuscript formatting, including the numbering of paragraphs and formulas, must be corrected 
  • The manuscript should have a uniform manner of captioning figures and linking captions directly to figures 
  • I suggest authors to unify legends where possible in order to make data easier to read, e.g. Fig. 3 and Fig. 1 
  • A rather unusual language is used, many of the vocabulary used is different than in the literature, e.g. to describe the progress of the process. 

2. Detailed comments 

Methodology: 

  • What rheometer measuring system was used? What was the value of the amplitude in oscillatory measurements? 
  • My doubts are raised by the value of the heating rate of 5°C/min. Why did the authors use this value? Such rapid heating can lead to significant errors due to the dynamics of the process. 

Results: 

  • Why do the authors use the phrase "hydration ratio" instead of "swelling ratio"? The latter is commonly used for hydrogels for biomedical applications. 
  • Figure 1: My doubts are raised by the peak for wavenumber 2979 for heparin. The shape of the peak is not typical and does not have a significant change in value. Did the authors compare their results with literature data? In the legend, missing the letter "P" in the PNIPA designation. 
  • Lines 305-311: The numerical values in the text are not consistent with the data presented in the figures, for example, the text gives 98% for heparin, and for the others even higher values. The graph shows that for heparin, values up to around 40 °C are even higher than 100%. It is further shown that one of the values decreased to 21% (line 309) - again this is not visible on the graph. 
  • What method of LCST determination did the authors use? The authors state that the LCST value was determined at the temperature at which solutions began to precipitate. If so, this should be the temperature point at which the transmittance value begins to decrease. Comparison of the values provided by the authors in text with the measurment points indicates no consistency in determining LCST. 
  • Lines 317-326: The authors indicate that the gelation point is the point at which the values of dynamic modules are equal.  There are other more accurate methods, e.g. Winter's or Frederickson-Larson's methods to determination the gelation temperature. The intersection point of the moduli is only a point of change in viscoelastic properties and cannot be identified with the formation of a fully devoped polymeric structure, given e.g. in Flory-Stockmayer's research. Moreover, the curves shape does not indicate the formation of a fully developed gel structure i.e. no plateau after phase transition. 
  • Figure 3b: The graph is illegible due to the large amount of experimental data. Can the authors replace it with the graph of damping factor? 
  • Figure 4b: data from the chart do not agree with the text, e.g. after 7 days the value should be 70% not 75%, after 8h approx. 15% not 40%. Especially the latter difference is significant. 
  • Line 353: in the text, the authors indicate the cells concentration used in the range of 0.1-1 while the graph presents the results of measurements in the range of 0.01-1 (+ control sample). The order of the concentrations in the figure 5a in unclear which may lead to a misunderstanding by the readers. 
  • Lines 385-386: The authors chose the system that has the highest LCST temperature (~37.5 °C, and ~36.9 °C from rheology) for in vivo testing. I have doubts about the kinetics of the gelation process, which may be different for studies on mice that have a higher body temperature than humans. I would consider using a system with a lower LCST temperature. I would like to suggest conducting phase transition studies at constant temperature (gelation kinetics). 
  • The title of the article contains "injection hydrogels", at the same time there is no research on the possible injection application of the systems under consideration. In this respect, I would suggest changing the title of the publication and replacing the "injectable", for example with "thermosensitive/thermoresponsive". 

Round 2

Reviewer 2 Report

The authors added some information about methodology of rhemoetric measurements i.e. the amplitude range (line 173). However this is totally different measurement, i.e. sweep amplitute test. The data was obtained from the measurements carried out under constant deformation (amplitude value). On the presented figures the intersection points are still not visible.